# Learning Transferable Adversarial Robust Representations via Multi-view Consistency

## Abstract

Despite the success on few-shot learning problems, most meta-learned models only focus on achieving good performance on clean examples and thus easily break down when given adversarially perturbed samples. While some recent works have shown that a combination of adversarial learning and meta-learning could enhance the robustness of a meta-learner against adversarial attacks, they fail to achieve generalizable adversarial robustness to unseen domains and tasks, which is the ultimate goal of meta-learning. To address this challenge, we propose a novel meta-adversarial multi-view representation learning framework with dual encoders. Specifically, we introduce the discrepancy across the two differently augmented samples of the same data instance by first updating the encoder parameters with them and further imposing a novel label-free adversarial attack to maximize their discrepancy. Then, we maximize the consistency across the views to learn transferable robust representations across domains and tasks. Through experimental validation on multiple benchmarks, we demonstrate the effectiveness of our framework on few-shot learning tasks from unseen domains, achieving over 10% robust accuracy improvements against previous adversarial meta-learning baselines.

## 1 Introduction

Recently proposed meta-learning approaches have shown impressive generalization ability to novel tasks while learning with few data instances (Koch et al., 2015; Sung et al., 2018; Snell et al., 2017; Finn et al., 2017), but are vulnerable to small imperceptible perturbations to the input data (Yin et al., 2018), i.e., adversarial attacks (Szegedy et al., 2014). To overcome such adversarial vulnerability of neural network-based meta-learners, several *adversarial meta-learning* (AML) (Yin et al., 2018; Goldblum et al., 2020; Wang et al., 2021) works have proposed to train robust meta-learners by combining class-wise attacks from adversarial training (AT) (Madry et al., 2018) with meta-learning methods (Finn et al., 2017; Raghu et al., 2019; Bertinetto et al., 2019). Previous AML approaches employ the *Adversarial Querying* mechanism (Goldblum et al., 2020; Wang et al., 2021) that meta-learns a shared initialization by taking an inner-adaptation step with the clean support set, while obtaining the adversarial robustness by AT on the query set at the outer optimization step.

Despite their successes, we find that the previous AML approaches (Figure 1a) are only effective in achieving adversarial robustness from seen domain tasks (e.g., CIFAR-FS, Mini-ImageNet), while showing poor transferable robustness to *unseen domains* (e.g., Tiered-ImageNet, CUB, Flower, Cars) as shown in Table 1. While the ultimate goal of meta-learning is obtaining transferable performance across various domain (Guo et al., 2020; Oh et al., 2022), which is a common occurrence in real-world, to the best of our knowledge, no research has yet targeted generalizable adversarial robustness in few-shot classification on unseen domains, leaving the problem largely unexplored.

In this paper, we posit that the vulnerability to domain shift in previous AML approaches arises from the adversarial training with *task-* and *domain-dependent* class-wise attacks. This integration inadvertently induces the adversarially overfitted robustness to a given domain and task type, focusing on learning robust decision boundaries for the few-shot classification tasks from the seen domain. This renders the learned decision boundaries ineffective when confronted with diverse tasks from unseen domains. To overcome this limitation, we leverage the efficacy of *self-supervised learning* (SSL) methods (Chen et al., 2020; He et al., 2020; Chen & He, 2021) in transferability such that the meta-learner aims to learn robust representations, rather than robust decision boundaries.

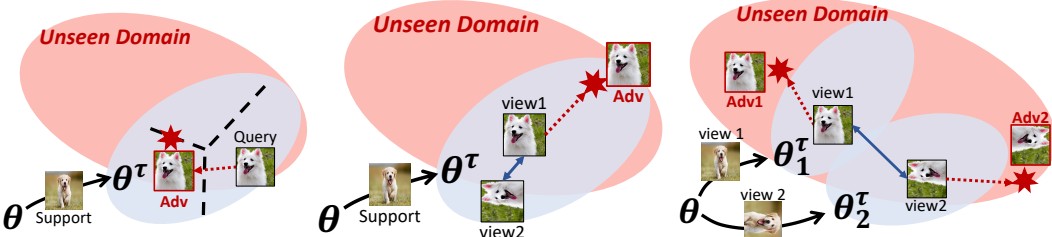

(a) Conventional AML methods (b) Naïve combination of SSL & AML    (c) Multi-view latent attack (Ours)

Figure 1: **Concept.** (a) Existing adversarial meta-learning (AML) method utilizes an adversarial querying mechanism that employs conventional class-wise attacks and cannot obtain robustness against adversaries from other domains. (b) Naïve combination of self-supervised learning (SSL) and AML is also suboptimal due to representational collapse with few data. (c) We introduce multi-view latent space to enrich the representation even with few data, and then apply label-free multi-view latent attacks to obtain maximal discrepancy across the views without representational collapse, and then learn adversarially robust representations.

Recent SSL demonstrates proficiency in acquiring transferable representations by learning view-invariant features by maximizing the similarity between the differently transformed instances of identical origin. This leads SSL to be able to attain strong structural recognition (Ericsson et al., 2021) based on large-scale data. However, since few-shot learning operates with a limited number of examples, a simple combination of SSL and AML approaches could not achieve adversarially robust representations for unseen domains due to adversarial representational collapse (Figure 1b, Table 6).

To overcome such limitation and leverage the transferability of SSL, we introduce a novel **M**eta-**A**dversarial multi-**v**iew **R**epresentation **L**earning (MAVRL) that explicitly minimizes the feature discrepancy between adversarial examples and clean image. MAVRL proposes **1) bootstrapped multi-view encoders** to obtain view-specialized latent spaces, which enriches the views even with the limited data, by taking an inner-gradient step from a shared encoder initialization using two distinct random augmentations applied on the same support set. We then introduce **2) label-free multi-view adversarial latent attacks**, which generate task-agnostic adversaries by maximizing the disagreement across different views in the shared latent space of our bootstrapped multi-view encoders (Figure 1c).

We extensively verify the efficacy of our proposed MAVRL against previous adversarial meta-learning methods (Yin et al., 2018; Goldblum et al., 2020; Wang et al., 2021) on multiple few-shot benchmarks. Notably, MAVRL improves both few-shot clean and robust accuracy against adversarial attack (Madry et al., 2018) on the unseen domains, from 32.49% → 50.32%, and 7.39% → 28.20% on average, respectively. To summarize, our contributions are as follows:

- We tackle a crucial problem of meta-adversarial learning, specifically the **transferability of the adversarial robustness across unseen tasks and domains with limited data**, which is an ultimate goal of the meta-learning for real-world application, yet has been unexplored in previous studies.

- We propose a **novel meta-adversarial framework, MAVRL**, which meta-learns transferable robust representations by minimizing the representational discrepancy across clean images and label-free multi-view latent adversarial examples.

- MAVRL obtains impressive generalized adversarial robustness on few-shot tasks from unseen domains. Notably, our model outperforms baselines by **more than 10%** in adversarial robust accuracy without compromising clean accuracy.

## 2 RELATED WORK

**Meta-learning.** Meta-learning (Thrun & Pratt, 1998) aims to learn general knowledge across a distribution of tasks in order to adapt quickly to new tasks with minimal data. There are two broad categories of meta-learning approaches: metric-based (Koch et al., 2015; Sung et al., 2018; Snell et al., 2017) and gradient-based (Finn et al., 2017; Nichol et al., 2018). In this work, we focus on gradient-based approaches which meta-learn a shared initialization (MAML (Finn et al., 2017)) or learning rate (Meta-SGD (Li et al., 2017)) using a bi-level optimization scheme consisting of inner- and outer-optimization steps. ANIL (Raghu et al., 2019) and BOIL (Oh et al., 2020) are two variations of gradient-based meta-learning; ANIL efficiently reuses features of the encoder

by updating only the classifier at the inner optimization step, in contrast, BOIL addresses domain shift by fixing the classifier and meta-learning the feature extractor. We chose the inner update rule of BOIL for ours since we aim at learning robust representations under domain shift. However, conventional meta-learning assumes the availability of abundant meta-training tasks, often impractical in real-world scenarios. Previous works (Wu et al., 2022; Yao et al., 2021; Liu et al., 2003) highlight the efficacy of task augmentation in enhancing generalized performance across unseen tasks by enriching the task distributions. Liu et al. (2003) involves image rotation at the input level, while at the representation level, Yao et al. (2021) incorporates a mix-up of cross-meta-training tasks and Wu et al. (2022) generates a new task by maximizing adversarial loss with additional up-sampling networks. Unlike existing approaches that augment tasks solely either at the input or representation level, ours enlarges the task distributions at both input and representation levels without additional networks. Our proposed bootstrapped encoders naturally augment tasks with the parameters of the encoders, generating more dissimilar tasks from their originals.

**Adversarial Meta-learning.** While meta-learning methods have shown promise in learning generalizable knowledge with limited data, yet they remain vulnerable to adversarial perturbations. To address this challenge, Yin et al. (2018) proposed to combine adversarial training with meta-learning. However, it is computationally expensive since AT is applied for both inner- and outer-optimization steps, and is further vulnerable to strong attacks. To overcome this, Adversarial Querying (AQ) (Goldblum et al., 2020) proposes to train with projected gradient descent (PGD) adversaries (Madry et al., 2018) only on the query set. RMAML (Wang et al., 2021) studies how to achieve robustness in a meta-learning framework and suggests a robustness-regularized meta-learner on top of the MAML. Despite their successes, we observe that previous adversarial meta-learning methods are vulnerable to *distributional domain shift* which often occurs in real-world applications. To overcome this limitation, we propose a novel meta-adversarial learning framework based on multi-view representation learning, which learns a transferable robust representation via label-free adversarial attacks along with consistency-based representation learning.

## 3 META-ADVERSARIAL MULTI-VIEW REPRESENTATION LEARNING

In this section, we introduce our proposed framework, *Meta-Adversarial Multi-view Representation Learning* (**MAVRL**). Before we describe the details of MAVRL, we first elaborate on our novel problem of learning transferable robust representations with limited data for meta-learners.

### 3.1 PROBLEM SETTING

Formally, we are interested in solving a few-shot classification task $\tau$ that consists of a train (support) set $\mathcal{S} = \{(x^s, y^s)\}_s$ and test (query) set $\mathcal{Q} = \{(x^q, y^q)\}_q$, where $x^s, x^q$ are input instances (e.g., images) and $y^s, y^q$ are their corresponding labels. The goal of conventional meta-learning (Finn et al., 2017) is to maximize the query set accuracy of a classifier trained with limited support set data for any unseen $N$-way $S$-shot task $\tau$. Thus, previous AML methods (Yin et al., 2018; Goldblum et al., 2020; Wang et al., 2021) demonstrated the accuracy on each task $\tau$ that is assumed to follow underlying task distribution $p_{\mathcal{D}}(\tau)$ associated with a *seen domain* $\mathcal{D}$ (e.g., Mini-ImageNet, CIFAR-FS). However, real-world tasks often extend beyond these seen domains, leading to a distributional domain shift where prior AMLs fail to obtain a robust meta-learner. To address this limitation on domain shift, we introduce a novel transferable adversarial robustness problem where meta-test tasks can be derived from any other task distribution, such as $p_{\mathcal{D}'}(\tau)$ associated with *unseen domain* $\mathcal{D}'$ (e.g., Tiered-ImageNet, CUB, Flower, Cars). Thus, our ultimate goal is to obtain an adversarially robust representation against any unseen tasks from any unseen domains, with limited data.

### 3.2 PRELIMINARY ON GRADIENT-BASED ADVERSARIAL META-LEARNING

Although there is a wide range of approaches for few-shot classification problems, we focus on the gradient-based meta-learning methods (Finn et al., 2017; Li et al., 2017; Raghu et al., 2019; Oh et al., 2020) due to their versatility. These approaches meta-learn a shared initialization of the neural network parameters (Finn et al., 2017) and element-wise inner learning rate (Li et al., 2017) with the bi-level optimization scheme, enabling rapid adaptation and generalization to unseen tasks by taking inner-gradient steps from the shared initialization. We now briefly review the recent adversarial meta-learning methods (Goldblum et al., 2020; Wang et al., 2021), which adopt class-wise adversarial

attacks built upon gradient-based meta-learning as follows:

$$\min_{\theta,\phi,\alpha} \mathbb{E}_{p_{\mathcal{D}}(\tau)} \Big[ \mathbb{E}_{\mathcal{Q}} \big[ \underbrace{\mathcal{L}_{\mathtt{ce}} \left( g_{\phi^\tau} \circ f_{\theta^\tau}(x^q), y^q \right)}_{\text{original meta-learning objective}} + \underbrace{\lambda \mathcal{L}_{\mathtt{kl}} \left( g_{\phi^\tau} \circ f_{\theta^\tau}((x^q)^{\mathtt{adv}}), g_{\phi^\tau} \circ f_{\theta^\tau}(x^q) \right)}_{\text{class-wise adversarial training}} \big] \Big],$$

$$\text{where } \underbrace{[\theta^\tau, \phi^\tau] = [\theta, \phi] - \alpha \odot \nabla_{\theta,\phi} \mathbb{E}_{\mathcal{S}} \left[ \mathcal{L}_{\mathtt{ce}} \left( g_\phi \circ f_\theta(x^s), y^s \right) \right]}_{\text{inner-gradient update}}.$$

(1)

Here, $\odot$ and $\circ$ denote element-wise product and composition of two functions, respectively. We assume that our model consists of feature encoder $f_\theta(\cdot)$ and classifier $g_\phi(\cdot)$ parameterized by $\theta$ and $\phi$, respectively. $\theta^\tau, \phi^\tau$ are adapted parameters by taking an inner-gradient step based on support set $\mathcal{S}$, where $\alpha, \lambda, \mathcal{L}_{\mathtt{ce}}(\cdot, \cdot)$, and $\mathcal{L}_{\mathtt{kl}}(\cdot, \cdot)$ are the element-wise inner learning rate, hyperparameter for balancing the trade-off between clean and robust accuracy, cross-entropy loss, and Kullback-Leibler divergence (KL) loss, respectively. For simplicity of notation, we only consider a single inner-gradient update here, but this can be straightforwardly extended to multiple gradient updates. For generality, we describe our meta-parameters as $\theta, \phi$, and $\alpha$, but it is common to optimize some of them for the AML literature (Yin et al., 2018; Goldblum et al., 2020; Wang et al., 2021). In meta-learning literature, there exist two variants that update only 1) the classifier parameter $\phi$ (Raghu et al., 2019), and 2) the encoder parameter $\theta$ (Oh et al., 2020) during inner-gradient steps. We adopt the second inner update rule (i.e., encoder only) on MAVRL since our focus is on learning representations.

The meta-level class-wise adversarial training in Eq. 1, dubbed as *adversarial querying* (AQ) mechanism (Goldblum et al., 2020; Wang et al., 2021), aims at learning to defend adversarial attack $(x^q)^{\mathtt{adv}}$ for each query $x^q$. Specifically, the adversarial example is the sum of query data and its adversarial perturbation, i.e., $(x^q)^{\mathtt{adv}} = x^q + \delta^q$, where the perturbation $\delta^q$ is generated to maximize the cross-entropy loss as follows:

$$\delta^q = \arg\max_{\delta \in B(x,\epsilon)} \mathcal{L}_{\mathtt{ce}}(g_{\phi^\tau} \circ f_{\theta^\tau}(x^q + \delta), y^q), \tag{2}$$

where $B(\cdot, \epsilon)$ is the $l_\infty$ norm-ball with radius $\epsilon$. Note that adversarial training of the AQ mechanism is only applied to outer optimization on the query set $\mathcal{Q}$, spurring two advantages: 1) cost-efficient adversarial robustness, and 2) superior clean performance on few-shot classification. Following the previous work, we employ adversarial training only at the outer optimization.

## 3.3 META-ADVERSARIAL REPRESENTATION LEARNING

Even though existing adversarial meta-learning methods (Yin et al., 2018; Goldblum et al., 2020; Wang et al., 2021) have shown to achieve clear improvements in adversarial robustness on few-shot classification tasks within the seen domain, we observe that they are highly vulnerable to domain shifts, i.e., $\mathcal{D} \to \mathcal{D}'$. We assume that the adversarial training on task- and domain-dependent class-wise attacks cause overfitting of adversarial robustness only on a seen domain $\mathcal{D}$. To alleviate domain-dependent adversarial robustness, we focus on learning transferable robust representations in a task-agnostic manner, which is motivated by *self-supervised learning* (SSL) (Chen et al., 2020; Chen & He, 2021). The recent learning objective of SSL is to minimize the distance between differently augmented views of the same image in the latent space based on pretext tasks generated from data.

**Naïve Adaptation of SSL on Meta-adversarial Training.** Our motivation is derived from self-supervised learning (SSL) that learns visual representation wherein augmented images coexist within the same latent space, which is label-free and effective in yielding transferable representation. One of the straightforward adaptations of SSL and AML is applying SSL-based meta-learning (Liu et al., 2021; Zhou et al., 2023) on the AML. However, simple employment of SSL could not contribute to the transferable adversarial robustness contrary to the success in achieving transferable clean performance in Table 3. Another trivial combination to obtain transferable robustness is adopting the self-supervised adversarial attack (Kim et al., 2020) on the query set $\mathcal{Q}$ (Figure 1 (b)) as follows:

$$\mathcal{L}_{\mathtt{sim}}(z, z_{\mathtt{pos}}, \{z_{\mathtt{neg}}\}) := -\log \frac{\exp(\mathtt{sim}(z, z_{\mathtt{pos}})/T)}{\exp(\mathtt{sim}(z, z_{\mathtt{pos}})/T) + \sum_{\{z_{\mathtt{neg}}\}} \exp(\mathtt{sim}(z, z_{\mathtt{neg}})/T)},$$

$$\delta^q = \arg\max_{\delta \in B(x,\epsilon)} \mathcal{L}_{\mathtt{sim}}(f_\theta(t_1(x^q) + \delta), f_\theta(t_2(x^q)), \{f_\theta(x^q_{\mathtt{neg}})\}),$$

(3)

---

**Algorithm 1:** **M**eta-**A**dversarial Multi-**v**iew **R**epresentation **L**earning (**MAVRL**).

---

**Input:** Meta-training distribution $p_{\mathcal{D}}(\tau)$, randomly selected data augmentations $t_1(\cdot), t_2(\cdot)$,
  feature encoder $f_{\theta}(\cdot)$, classifier $g_{\phi}(\cdot)$, meta-learning rate $\beta$
**Output:** Adversarially meta-trained parameters $\theta, \phi, \alpha$
**while** *not converged* **do**
  Sample $M$ different meta-training tasks $\{\tau\} = \{(\mathcal{S}, \mathcal{Q})\} \sim p_{\mathcal{D}}(\tau)$
  **for** $i = 1, \cdots, M$ **do**
    ```
    /* Bootstrap multi-view encoders with support set S.          */
    ```
    $\theta_j^{\tau} \leftarrow \theta - \alpha \odot \nabla_{\theta}\mathbb{E}_{\mathcal{S}}[\mathcal{L}_{\text{ce}}(g_{\phi} \circ f_{\theta}(t_j(x^s)), y^s)], \text{for } j = 1, 2$ `// Details in Eq. 4`

    ```
    /* Generate multi-view latent adversaries using query set Q.  */
    ```
    $t_j(x^q)^{\text{adv}} = t_j(x^q) + \delta_j^q, \text{for } j = 1, 2$         `// `$\delta_j^q$` are obtained by Eq. 5`

    ```
    /* Compute meta-adversarial multi-view representation learning
       loss for a given task τ.                                   */
    ```
    $\mathcal{L}_{\text{ours}}^{\tau} = \mathbb{E}_{\mathcal{Q}}\big[\sum_{j=1,2}\big(\mathcal{L}_{\text{ce}}(\cdot,\cdot) + \lambda\mathcal{L}_{\text{kl}}(\cdot,\cdot)\big) + \mathcal{L}_{\text{cos}}(\cdot,\cdot)\big]$    `// Details in Eq. 6`

  ```
  /* Update our meta-parameters using gradient descent algorithms  */
  ```
  $[\theta, \phi, \alpha] \leftarrow [\theta, \phi, \alpha] - \beta\nabla_{\theta,\phi,\alpha}\sum_{\{\tau\}}\mathcal{L}_{\text{ours}}^{\tau}/M$
**return** meta-parameters $\theta, \phi, \alpha$

---

where $t_1(\cdot), t_2(\cdot)$ are two randomly selected data augmentations to a given batch $\{x\}$ and define $x_{\text{pos}}$ of $t_1(x)$ as $t_2(x)$. The remaining instances in the batch $\{x\}$ are then defined as $\{x_{\text{neg}}\}$. $z, z_{\text{pos}}$, and $\{z_{\text{neg}}\}$ are latent vectors obtained from the feature encoder $f_{\theta}(\cdot)$. The $\text{sim}(\cdot,\cdot)$ and $T$ are cosine similarity functions and a temperature term, respectively. However, the trivial combination of self-supervised adversarial attack to AML could not ensure the transferable adversarial robustness of meta-learners, as shown in Table 3. We attribute the failure of simple modification to the well-known problem of the contrastive objective in a small batch, *representational collapse* (Chen & He, 2021; Zbontar et al., 2021) of adversarial examples: models trivially produce similar or even identical representations for different adversarial examples, especially when using small batch sizes in self-supervised adversarial attack. Conventional few-shot learning settings (e.g., $|\mathcal{S}| = 5 \times 5, |\mathcal{Q}| = 5 \times 15$), by their own definition, suffer from the severe representational collapse between different adversarial examples, leading adversaries to be ineffective. This hinders meta-learners from achieving both generalized adversarial robustness and clean performance in unseen domains.

**Bootstrapping Multi-view Encoders from Meta-learner.** To overcome the above challenge, we propose a novel scheme to enhance the representation power even within a few data by introducing *bootstrapped view-specialized feature encoders*. Bootstrapped multi-view encoders are obtained by taking inner-gradient steps from the meta-initialized ($\theta$) encoder with two views of differently augmented support set images $\mathcal{S}$ as follows:

$$\theta_1^{\tau} \leftarrow \theta - \alpha \odot \nabla_{\theta}\mathbb{E}_{\mathcal{S}}[\mathcal{L}_{\text{ce}}(g_{\phi} \circ f_{\theta}(t_1(x^s)), y^s)],$$
$$\theta_2^{\tau} \leftarrow \theta - \alpha \odot \nabla_{\theta}\mathbb{E}_{\mathcal{S}}[\mathcal{L}_{\text{ce}}(g_{\phi} \circ f_{\theta}(t_2(x^s)), y^s)],$$
$$(4)$$

where $t_1, t_2$ is the stochastic data augmentation functions, including random crop, random flip, random color distortion, and random grayscale as Zbontar et al. (2021). Our view-specialized feature encoders generate multi-view parameter space on top of each augmented input space, inducing the representation space to be enlarged. This expansion enhances the exploratory capacity of self-supervised adversarial attacks, leading to a more extensive investigation of unseen domains as shown in Figure 1 (c) and further mitigates the adversarial representational collapse. Unlike recent self-supervised learning methods (He et al., 2020; Chen & He, 2021), which use stop-gradient or momentum network to generate multi-view representations within the same parameter space, our approach utilizes multi-view parameter space obtained from bootstrapped multi-view encoders. Thus, our proposed encoders operate with a representation that is expanded twofold by employing meta-learning specialized bootstrapped parameters. This fundamental difference amplifies the effectiveness of our method on adversarial robustness in transferable few-shot classification.

**Multi-view Adversarial Latent Attacks.** On top of the proposed bootstrapped multi-view encoders, our novel multi-view adversarial latent attacks generate perturbations by maximizing the discrepancy across the latent features obtained from the bootstrapped multi-view encoders $f_{\theta_1^\tau}(\cdot), f_{\theta_2^\tau}(\cdot)$ through the iterative algorithm, projected gradient descent (Madry et al., 2018), as follows:

$$\delta_1^{i+1} = \prod_{B(x,\epsilon)}\Big(\delta_1^i + \gamma\,\mathtt{sign}\big(\nabla_{\delta_1^i}\mathcal{L}_{\mathtt{sim}}(f_{\theta_1^\tau}(t_1(x^q)+\delta_1^i), f_{\theta_1^\tau}(t_2(x^q)), \{f_{\theta_1^\tau}(x_{\mathtt{neg}}^q), f_{\theta_2^\tau}(x_{\mathtt{neg}}^q)\})\big)\Big),$$

$$\delta_2^{i+1} = \prod_{B(x,\epsilon)}\Big(\delta_2^i + \gamma\,\mathtt{sign}\big(\nabla_{\delta_2^i}\mathcal{L}_{\mathtt{sim}}(f_{\theta_2^\tau}(t_2(x^q)+\delta_2^i), f_{\theta_2^\tau}(t_1(x^q)), \{f_{\theta_1^\tau}(x_{\mathtt{neg}}^q), f_{\theta_2^\tau}(x_{\mathtt{neg}}^q)\})\big)\Big),$$

(5)

where $\delta_1^i, \delta_2^i$ are generated perturbations for each view with $i$ attack steps, $\gamma$ step size of the attack, and attack objective of $\mathcal{L}_{\mathtt{sim}}(\cdot,\cdot,\cdot)$ which is the contrastive loss in Eq. 3. The final adversarial examples are then obtained by adding the perturbations to each transformed image, i.e., $t_1(x^q)^{\mathtt{adv}} = t_1(x^q) + \delta_1^i, t_2(x^q)^{\mathtt{adv}} = t_2(x^q) + \delta_2^i$.

**Robust Representation Learning with Multi-view Consistency.** Building upon the proposed multi-view adversarial latent attacks, we introduce our multi-view adversarial meta-learning method that explicitly aims to learn transferable robust representations along with view consistency. Formally, our meta-objective is defined as follows:

$$\min_{\theta,\phi,\alpha} \mathbb{E}_{p_\mathcal{D}(\tau)}\Big[\mathbb{E}_\mathcal{Q}\Big[\underbrace{\sum_{j=1,2}\Big(\overbrace{\mathcal{L}_{\mathtt{ce}}(g_\phi \circ f_{\theta_j^\tau}(t_j(x^q)), y^q)}^{\text{original meta-learning objective}} + \lambda\mathcal{L}_{\mathtt{kl}}(g_\phi \circ f_{\theta_j^\tau}(t_j(x^q)^{\mathtt{adv}}), g_\phi \circ f_{\theta_j^\tau}(t_j(x^q)))\Big)}_{\text{multi-view adversarial training}}$$

$$+ \underbrace{\mathcal{L}_{\mathtt{cos}}(f_{\theta_1^\tau}(t_1(x^q)^{\mathtt{adv}}), f_{\theta_2^\tau}(t_2(x^q)^{\mathtt{adv}}))}_{\text{multi-view consistency}}\Big]\Big],$$

$$\text{where}\quad \theta_j^\tau = \underbrace{\theta - \alpha \odot \nabla_\theta \mathbb{E}_\mathcal{S}\big[\mathcal{L}_{\mathtt{ce}}\big(g_\phi \circ f_\theta(t_j(x^s)), y^s\big)\big]}_{\text{bootstrapped multi-view encoders}}.$$

(6)

We meta-learn the shared initialization of the encoder parameter $\theta$, classifier parameter $\phi$, and inner learning rate $\alpha$ (Li et al., 2017). Our meta-objective consisting of cross-entropy loss, multi-view adversarial loss, and multi-view consistency loss are computed over both bootstrapped encoders. $\mathcal{L}_{\mathtt{cos}}(\cdot,\cdot)$ is cosine distance loss, i.e., $1 - (x^\mathsf{T}y)/(\|x\|\|y\|)$, which minimizes feature representations between the multi-view adversaries, enforcing the multi-view consistency to enable our meta-learner explicitly to learn robust representations that are invariant to views. We present an overall meta-adversarial multi-view representation learning in Algorithm 1, a meta-leaner can learn transferable adversarial robust representations for unseen tasks and domains, by learning consistency-based representation between the label-free adversaries from the different views.

## 4 EXPERIMENT

In this section, we introduce the experimental setup (Section 4.1) and validate MAVRL's adversarial robustness on novel few-shot learning tasks from unseen domains (Section 4.2). We then conduct ablation experiments to analyze the proposed components (Section 4.3).

### 4.1 EXPERIMENTAL SETUP.

**Datasets.** For meta-training, we use CIFAR-FS (Bertinetto et al., 2019) and Mini-ImageNet (Russakovsky et al., 2015). We validate meta-learners on six few-shot classification benchmarks for adversarial robust transferability: CIFAR-FS (Bertinetto et al., 2019), Mini-ImageNet (Russakovsky et al., 2015), Tiered-ImageNet (Russakovsky et al., 2015), Cars (Krause et al., 2013), CUB (Welinder et al., 2010) and Flower (Nilsback & Zisserman, 2008).

**Baselines.** We consider clean meta-learning (CML) and existing adversarial meta-learning (AML) methods as our baselines. **1) MAML** (Finn et al., 2017): The gradient-based clean meta-learning method *without adversarial training*. **2) MetaOptNet** (Lee et al., 2019): The metric-based clean meta-learning method. **3) ADML** (Yin et al., 2018): The simple combination of adversarial training on both inner and outer optimization. **4) AQ** (Goldblum et al., 2020): The adversarial querying (AQ) mechanism in Eq. 1, where only adversarial training in outer optimization with the differentiable analytic solver (Bertinetto et al., 2019). **5) RMAML** (Wang et al., 2021): The exact AQ mechanism in Eq. 1, except for several gradient steps in the inner optimization.

Table 1: Results of **adversarial robustness** for 5-way 5-shot classification tasks on unseen and seen domains. All adversarial meta-learning methods are trained on CIFAR-FS or Mini-ImageNet. CML stands for the clean meta-learning. AML stands for adversarial meta-learning. Rob. stands for accuracy (%) calculated with PGD-20 attack ($\epsilon = 8./255., \gamma = \epsilon/10$). **Bold** and underline stands for the best and second.

| Type | CIFAR-FS → | Mini-ImageNet | | Tiered-ImageNet | | CUB | | Flower | | Cars | | Avg. | | CIFAR-FS | |
|---|---|---|---|---|---|---|---|---|---|---|---|---|---|---|---|
| | | Clean | Rob. | Clean | Rob. | Clean | Rob. | Clean | Rob. | Clean | Rob. | Clean | Rob. | Clean | Rob. |
| CML | MAML (Finn et al., 2017) | 44.85 | 6.21 | **61.19** | 2.48 | 48.41 | 3.46 | **67.76** | 5.73 | 43.94 | 5.31 | **53.83** | 4.24 | 75.10 | 12.20 |
| | MetaOptNet (Lee et al., 2019) | 34.93 | 0.02 | 37.07 | 0.00 | 45.52 | 0.00 | 65.92 | 0.00 | 45.22 | 0.00 | 45.73 | 0.00 | **80.95** | 0.00 |
| AML | ADML (Yin et al., 2018) | 28.66 | 6.53 | 40.06 | 11.36 | 31.18 | 5.21 | 39.36 | 11.26 | 27.43 | 3.18 | 33.34 | 7.10 | 53.06 | 22.45 |
| | AQ (Goldblum et al., 2020) | 33.09 | 3.32 | 37.41 | 5.05 | 38.37 | 4.10 | 60.14 | 11.03 | 36.83 | 4.47 | 41.96 | 5.99 | 73.19 | 42.82 |
| | RMAML (Wang et al., 2021) | 28.05 | 6.65 | 29.54 | 9.30 | 30.24 | 5.67 | 42.91 | 10.79 | 31.72 | 5.56 | 32.49 | 7.39 | 57.95 | 35.30 |
| | Ours | **45.82** | **24.12** | 51.46 | **30.06** | **48.56** | **25.23** | 66.49 | **42.16** | 38.29 | **19.43** | 50.32 | **28.20** | 67.75 | **43.42** |

| Type | Mini-ImageNet → | CIFAR-FS | | Tiered-ImageNet | | CUB | | Flower | | Cars | | Avg. | | Mini-ImageNet | |
|---|---|---|---|---|---|---|---|---|---|---|---|---|---|---|---|
| | | Clean | Rob. | Clean | Rob. | Clean | Rob. | Clean | Rob. | Clean | Rob. | Clean | Rob. | Clean | Rob. |
| CML | MAML (Finn et al., 2017) | 66.75 | 12.97 | **65.33** | 13.10 | 52.82 | 4.46 | **71.01** | 4.86 | **43.66** | 2.77 | **59.31** | 7.23 | **58.51** | 5.26 |
| | MetaOptNet (Lee et al., 2019) | **70.12** | 0.00 | 43.78 | 0.00 | 47.39 | 0.00 | 62.77 | 0.00 | 37.97 | 0.00 | 52.41 | 0.00 | 40.57 | 0.00 |
| AML | ADML (Yin et al., 2018) | 41.14 | 13.36 | 41.05 | 13.26 | 32.82 | 4.59 | 43.07 | 9.65 | 24.85 | 5.48 | 36.79 | 9.46 | 26.72 | 6.81 |
| | AQ (Goldblum et al., 2020) | 61.97 | 30.73 | 47.61 | 14.21 | 45.64 | 13.19 | 65.40 | 25.01 | 37.29 | 8.85 | 51.18 | 18.80 | 36.72 | **22.89** |
| | RMAML (Wang et al., 2021) | 37.94 | 10.59 | 30.49 | 8.24 | 27.30 | 6.26 | 42.52 | 13.08 | 37.76 | 5.43 | 35.20 | 8.92 | 43.98 | 21.47 |
| | Ours | 65.45 | **36.51** | 59.64 | **29.73** | **53.70** | **20.64** | 69.84 | **36.49** | 42.25 | **14.42** | 58.37 | **27.96** | 47.56 | 18.18 |

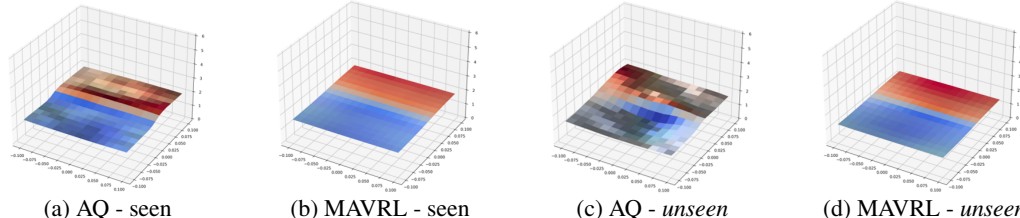

(a) AQ - seen        (b) MAVRL - seen        (c) AQ - *unseen*        (d) MAVRL - *unseen*

Figure 2: Loss surfaces for **(a)**, **(b)**: the seen domain (CIFAR-FS); **(c)**, **(d)**: the *unseen* domain (Mini-ImageNet).

**Implementational Details.** For all methods including ours, ResNet-12 is used as a backbone encoder $f_\theta(\cdot)$ and fully-connected layer is used as a classifier $g_\pi(\cdot)$. We consider the following conventional few-shot learning settings: 5-way 5-shot support set images and 5-way 15-shot query set images. For the meta-test, the meta-learners are evaluated with 400 randomly selected tasks. Note that our method takes just a single step for the inner optimization of meta-training and meta-test for computational efficiency. For adversarial training, we takes $i = 7$ gradient steps with the $\ell_\infty$ norm ball size $\epsilon = 8.0/255.0$ and the step size $\gamma = 2.0/255.0$. We set the regularization hyperparameter of TRADES (Zhang et al., 2019), i.e., $\lambda$, as 6.0. The adversarial robustness is evaluated against PGD (Madry et al., 2018) attacks by taking $i = 20$ gradient steps with the $\ell_\infty$ norm ball size $\epsilon = 8.0/255.0$ and the steps size $\gamma = 8.0/2555.0$. For baselines, we follow the original paper to set hyperparameters, such as the number of inner-steps, or inner learning rate. More experimental details are described in Supplementary A.

## 4.2 EXPERIMENTAL RESULTS ON FEW-SHOT TASKS

**Adversarial Robustness on Unseen Domain Tasks.** Given that our main goal is to attain transferable robustness on tasks from unseen domains, we mainly validate our method on unseen domain few-shot classification tasks. We meta-train MAVRL on CIFAR-FS (or Mini-ImageNet) and meta-test it on other benchmark datasets from different domains such as Mini-ImageNet (or CIFAR-FS), Tiered-ImageNet, CUB, Flower, and Cars. As shown in Table 1, MAVRL achieves impressive transferable robustness on unseen domain tasks, while previous AML methods easily break down to adversarial attacks from the unseen domains. In particular, MAVRL outperforms the baselines by more than 10% in robust accuracy even though the distribution of the unseen domains (i.e., CUB, Flower, and Cars) which are different from the distribution of the meta-trained dataset.

To demonstrate the transferability of MAVRL to significant domain shifts, i.e., non-RGB domains, we evaluate adversarial robustness on EuroSAT (Helber et al., 2019), ISIC (Codella et al., 2018), and CropDisease (Mohanty et al., 2016) (Table 2). Notably, the result highlights that MAVRL obtains outstanding adversarial robust accuracy when facing substantial domain shifts, surpassing the state-of-the-art AML method.

Table 2: Transferable adversarial robustness in non-RGB unseen domain tasks that are trained on CIFAR-FS.

| CIFAR-FS → | EuroSAT | | ISIC | | CropDisease | | Avg. | |
|---|---|---|---|---|---|---|---|---|
| | Clean | Rob. | Clean | Rob. | Clean | Rob. | Clean | Rob. |
| ADML | 60.41 | 7.79 | **32.20** | 0.51 | 50.15 | 1.64 | 47.59 | 3.31 |
| RMAML | 52.21 | 6.56 | 29.23 | 2.46 | 55.08 | 5.43 | 45.51 | 4.82 |
| AQ | 46.05 | 4.62 | 31.90 | 0.62 | 47.38 | 0.51 | 41.78 | 1.92 |
| Ours | **59.39** | **19.90** | 30.77 | **5.23** | **57.85** | **27.28** | **49.34** | **17.47** |

Table 3: Results of ablation experiments of naïve combination of previous meta-learning, self-supervised learning (SSL), and adversarial training approaches. All adversarial meta-learning methods are trained on CIFAR-FS. Rob. stands for accuracy (%) calculated with PGD-20 attack ($\epsilon = 8.255.$, $\gamma = \epsilon 10$).

| Naïve Combination Ablation | | | Mini-ImageNet | | Tiered-ImageNet | | CUB | | Flower | | Cars | | Avg. | |
|---|---|---|---|---|---|---|---|---|---|---|---|---|---|---|
| Meta-learning | SSL | Adversarial Training | Clean | Rob. | Clean | Rob. | Clean | Rob. | Clean | Rob. | Clean | Rob. | Clean | Rob. |
| MAML (Finn et al., 2017) | - | AT (Madry et al., 2018) | 28.66 | 6.53 | 40.06 | 11.36 | 31.18 | 5.21 | 39.36 | 11.26 | 27.43 | 3.18 | 33.34 | 7.10 |
| ProtoNet (Snell et al., 2017) | - | AT (Madry et al., 2018) | 33.19 | 2.61 | 37.15 | 4.13 | 36.56 | 3.82 | 62.43 | 13.38 | 40.45 | 4.46 | 41.96 | 5.68 |
| MetaOptNet (Lee et al., 2019) | - | AT (Madry et al., 2018) | 35.02 | 5.41 | 39.09 | 8.71 | 44.19 | 9.75 | 69.07 | 28.99 | 40.32 | 10.21 | 45.54 | 12.61 |
| infoPatch (Liu et al., 2021) | | AT (Madry et al., 2018) | 66.28 | 9.44 | 68.78 | 12.32 | 47.99 | 3.90 | 78.89 | 20.23 | 62.80 | 6.20 | **64.94** | 10.42 |
| LDP (Zhou et al., 2023) | | AT (Madry et al., 2018) | 32.55 | 12.33 | 38.93 | 18.10 | 34.50 | 10.60 | 52.46 | 24.01 | 37.25 | 14.50 | 39.14 | 15.91 |
| MetaSGD (Li et al., 2017) | | RoCL (Kim et al., 2020) | 20.30 | 17.99 | 21.70 | 18.66 | 21.59 | 18.19 | 24.77 | 21.33 | 21.74 | 19.30 | 22.02 | 19.09 |
| Ours | | | 45.82 | 24.12 | 51.46 | 30.06 | 48.56 | 25.23 | 66.49 | 42.16 | 38.29 | 19.43 | 50.32 | **28.20** |

We remark that in the absence of adversarial training, clean meta-learning (CML) approaches consistently failed to achieve any measure of adversarial robustness, regardless of whether the domain was seen or unseen. This observation underscores the extreme vulnerability of CML, showing a decrease in accuracy to approximately 10% against adversarial attacks. This vulnerability emphasizes the significant importance of robust training against attackers when dealing with potential threats in CML. While achieving adversarial robustness is necessary, previous AML works sacrifice clean accuracy to obtain adversarial robustness, especially in unseen domains. Contrarily, MAVRL can preserve clean accuracy akin to the CML method in unseen domains, while exhibiting outstanding adversarial robustness. This ability stems from the focus of MAVRL on learning robust representations, rather than focusing solely on learning robust decision boundaries.

**Visualization on Loss Surfaces and Representation Space of Adversaries.** To explore how MAVRL can obtain adversarial robustness on unseen domains, we first visualize the cross-entropy loss surface of an image by adding noise (Li et al., 2018). The loss surface indicates the ability of the model to generate consistent outputs even when subjected to a wide range of small noise on the input. As shown in Figure 2, both AQ and MAVRL exhibit relatively smooth loss surfaces when operating on a seen domain (CIFAR-FS), allowing both models to be robust to small adversarial perturbations. However, AQ shows a rough loss landscape in the unseen domain, indicative of the relative dissimilarity between the output spaces of attacked and clean images. Conversely, MAVRL has a smoother loss surface, demonstrating its ability to extract perturbation-invariant features in unseen domains, thereby leading to better transferable robustness.

To verify whether the model can capture distinctive visual features in any unseen domain, we visualize the representation space of an unseen domain, CIFAR-10, using t-SNE (Van der Maaten & Hinton, 2008). Figure 3 shows that MAVRL is able to obtain a well-separated feature space for adversarial examples in this novel domain. In contrast, AQ presents a substantially overlapped feature space across adversarial instances belonging to different classes, indicated by red dots scattered on diverse clusters. This observation suggests that the superior adversarial robustness of MAVRL in

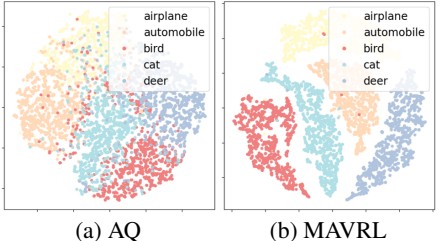

(a) AQ      (b) MAVRL

Figure 3: Representation visualization on the unseen domain, CIFAR-10.

unseen domains in Table 1 mainly stems from its ability to extract robust visual features from the input of any domain through the proposed multi-view meta-adversarial representation learning.

## 4.3 ABLATION STUDY

We now conduct an extensive ablation study to verify 1) why a simple combination of SSL and adversarial meta-learning could not achieve comparable performance as our multi-view meta-adversarial representation learning (MAVRL) and 2) the effectiveness of each proposed component.

**A Naïve Combination of Adversarial Meta-learning with SSL Cannot Achieve Generalizable Adversarial Robustness.** Our method provides a novel attack scheme that maximizes the representational discrepancy across the views, along with a consistency-based robust representation learning scheme, which is not a mere combination of adversarial learning, SSL, and meta-learning. A naïve combination of these strategies does not yield the same level of transferable adversarial robustness, as evidenced in Table 3. Combining meta-learning and adversarial training (Madry et al., 2018) fails to provide transferable performance in both clean and robust settings, as shown in previous AML approaches. While combining self-supervised learning-based meta-learning (Liu et al., 2021; Zhou et al., 2023) with class-wise adversarial training grants transferable clean performance, it fails to

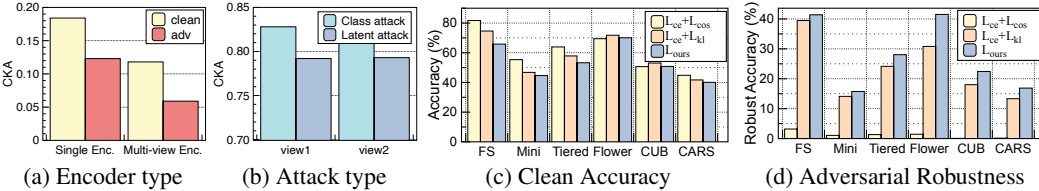

Figure 4: Ablation studies on multi-view components: **(a)**, **(b)**, and meta-learning objectives: **(c)**, **(d)**.

obtain adversarial robustness. Additionally, employing self-supervised adversarial training (Kim et al., 2020) within meta-learning forces a compromise on clean performance for transferable adversarial robustness. Contrarily, our bootstrapped multi-view representation learning successfully delivers exceptional clean and adversarial robustness in unseen domains.

**Bootstrapped Multi-view Encoders Contribute to Enlarged Representation Space.** To demonstrate the efficacy of the multi-view encoders, we train the MAVRL without bootstrapping, using only *a single encoder* for inner-adaptation while employing the same augmentations. As

Table 4: Ablation study of multi-view encoders in the inner-adaptation step on unseen domains.

| Encoder Type | Mini-ImageNet | | Tiered-ImageNet | | Flower | | Cars | | CUB | |
|---|---|---|---|---|---|---|---|---|---|---|
| | Clean | Rob. | Clean | Rob. | Clean | Rob. | Clean | Rob. | Clean | Rob. |
| Single | 40.39 | **17.04** | 52.65 | 27.33 | 67.92 | 37.58 | 39.03 | 16.11 | 49.06 | 21.25 |
| Multi-view | **44.64** | 15.75 | **53.25** | **28.05** | **70.08** | **41.52** | **40.08** | **16.88** | **50.78** | **22.44** |

shown in Table 4, our bootstrapping mechanism leads to a substantial improvement in both clean and robust accuracy on unseen domain tasks, as opposed to the MAVRL trained with a single encoder. This suggests that integrating only multi-view adversarial latent attack and multi-view consistency into meta-learning methods is not advantageous to learning transferable robust representations.

To examine the representational discrepancy introduced by bootstrapped multi-view encoders, we measure the Centered Kernel Alignment (CKA) (Kornblith et al., 2019) value between two views in the latent space for CIFAR-FS. CKA value is high when the two distributions are more similar and is 1 when two distributions are exactly the same. Figure 4a shows that features of two views from the bootstrapped multi-view encoders are more dissimilar (CKA ↓) than that from the single encoder (CKA ↑). These results support that our multi-view encoders contribute to producing distinct latent vectors for the same instance and enlarging the representation space.

**Multi-view Latent Attacks Make Stronger Attacks.** We further analyze the effectiveness of our multi-view latent attack within MAVRL compared to a class-wise attack. The distinction lies solely in the attack loss used to generate adversarial pertur-

Table 5: Ablation study of attack type on unseen domains.

| Attack Type | Mini-ImageNet | | Tiered-ImageNet | | Flower | | Cars | | CUB | |
|---|---|---|---|---|---|---|---|---|---|---|
| | Clean | Rob. | Clean | Rob. | Clean | Rob. | Clean | Rob. | Clean | Rob. |
| Class-wise | 42.15 | **17.13** | **53.91** | 27.41 | 69.66 | 38.83 | 40.05 | 16.37 | 50.01 | 21.20 |
| Multi-view latent | **44.64** | 15.75 | 53.25 | **28.05** | **70.08** | **41.52** | **40.08** | **16.88** | **50.78** | **22.44** |

bations. Class-wise attacks maximize cross-entropy loss with task labels, while our attacks maximize contrastive loss between multi-view latent vectors as in Eq. 5. Table 5 shows that the meta-learner trained with our multi-view latent attacks consistently shows better adversarial robustness than the class-wise attacks. This is because while the class-wise attack generates adversarial examples by only crossing the decision boundary of the seen domain task, the multi-view latent attack creates adversarial examples in any direction that is from the original image in the latent space, even with limited data. This implies that the multi-view latent attack has a larger attack range, enabling the use of stronger adversarial examples for a more robust representation. To support this, we report the CKA between clean and adversarial features generated by class-wise attacks and multi-view latent attacks, respectively. As shown in Figure 4b, multi-view latent attack produces more distinct, i.e., difficult, adversarial examples which are highly dissimilar from those of the clean images (CKA ↓).

**Multi-view Consistency Loss Regularized to Learn Generalized Features.** Our meta-objective consists of two terms: multi-view adversarial training loss $\mathcal{L}_{ce}(\cdot, \cdot) + \lambda\mathcal{L}_{kl}(\cdot, \cdot)$, and multi-view consistency loss $\mathcal{L}_{cos}(\cdot, \cdot)$ as in Eq. 6. The adversarial training loss is calculated on the logit independently on each view to enhance the robustness of each training sample. On the other hand, the consistency loss is computed with cosine distance loss between the features obtained from the bootstrapped encoders, enforcing consistency across view-specialized features generated from a multi-view latent attack. Thus, the model can learn a consistent representation of adversarial examples across tasks. In Figure 4c and 4d, we validate each term by conducting an ablation experiment using meta-learners trained on CIFAR-FS. Notably, we observe that adversarial robustness in unseen domains is significantly improved with the proposed multi-view consistency loss, which demonstrates that the view-invariant consistency contributes to transferable adversarial robustness.

## 5   CONCLUSION

In this paper, we address the important and yet unexplored problem of adversarial meta-learning under domain-shifted realistic scenarios. The focus is on ensuring adversarial robustness in the meta-learner over unseen tasks and domains with limited data. To tackle this challenge, we propose a novel meta-adversarial multi-view representation learning framework which is comprised of three components: 1) bootstrapped multi-view encoders that expand the representation space by generating multi-view parameter space on top of each view at the inner-adaptation; 2) label-free multi-view latent attacks generate stronger adversarial examples that mitigate adversarial representation collapse; and 3) multi-view consistency objectives between views to learn view-consistent visual representations, for enhanced transferability. Experimental results confirm that our model achieves outstanding transferable adversarial robustness on few-shot learning tasks from unseen domains.

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
