# OpenReview forum: "Learning Transferable Robust Representations for Few-shot Learning via Multi-view Consistency"
_ICLR.cc/2024/Conference — Submitted to ICLR 2024_

### Official Review · Reviewer_fCDy · 2023-10-28

**Soundness:** 2 fair
**Presentation:** 3 good
**Contribution:** 1 poor
**Rating:** 3
**Confidence:** 5

**Summary:**

This work aims to improve the robustness of meta-learning methods under domain shift. To this end, the authors introduce contrastive learning to the adversarial meta-training process. They propose to bootstrap multi-view encoders instead of single one to overcome representation collapse. The experiments on CIFAR-FS and mini-ImageNet verify the effectiveness of proposed method.

**Strengths:**

The paper is organized and written well and the paper looks quite well polished. The overall story is clear to me.

**Weaknesses:**

My main concerns are the meaning of the problem solved in this work and effectiveness of the proposed method. Concretely,
1. I do not think it is still mearningful for few-shot learning to meta-train on a small base dataset nowadays. In fact, since the pre-trained large vision bockbone, like CLIP and MoCo, have been provided and have far more generalizing representation than the model trained on small base dataset, I think it is more useful to use them for few-shot learning. It can hindle many different target domains simultaneously and also obtain very strong performance, as shown in [a,b].

2. Even compared with existing models under traditional setting (training on base dataset), the performance of the proposed method is far lag behind them in clean setting. For example, for 5-shot tasks, FeLMi [c] achieves 89.47 on CIFAR-FS, 86.08 on miniImageNet and 77.61 on miniImageNet-->CUB, while this work only obtains 67.75, 47.56 and 53.70 respectively. Although the performance is improved in robust setting, the cost in clean setting is too expensive, which makes it meaningless.

3. Considering the robust test is based on PGD attack and the model also use PGD attack for training, so the main reason why this method works under the robust setting is the use of project gradient descent (PGD), which is already a very common method.

4. It is hard to believe that MetaOptNet performs worse than MAML in both clean and robust setting. In my experience, although MAML has a wider range of applications, such as reinforcement learning and so on, MAML is lag behind MetaOptNet in terms of few-shot classification. A direct evidence is the Table 6 and Table 12 in [d], where MetaOptNet achieves far better performance than MAML in both clean and robust setting. I think they are the reasonable results. Even we evaluate in the cross-domain settings, the results shouldn't be so different, since mini/Tiered-ImageNet or CARS are not so different with CIFAR-FS. MetaOptNet still should outperform MAML, as least in clean setting, although the difference could be smaller than in-domain.

a. Learning to prompt for vision-language models.
b. PLOT: Prompt learning with optimal transport for vision-language models.
c. FeLMi: few shot learning with hard mixup.
d. Adversarially Robust Few-Shot Learning: A Meta-Learning Approach.

**Questions:**

Please answer the questions in weakness.

---

> ### Author Response · Authors · 2023-11-16
> **Response to your comments (1/2)**
>
> **Weakness 1**: Few-shot learning to meta-train on small-based datasets is no longer meaningful research.
> * We respectfully contest this claim as it overlooks the relevance of such research in scenarios with computational or data constraints.
> * While we recognize the impressive performance of large vision models like CLIP and MoCo, it is important to note that not all real-world problems afford the luxury of extensive computational resources and large datasets. For example, in the clinic problem or individual personal services, one might only have a limited number of samples for each class for diagnosis prediction or personal recognition and simultaneously need to safeguard against the threats of attacks.
> * Consequently, a large number of data for each class is not a universally applicable or feasible approach for all real-world applications.
> * Existing works [1, 2] also emphasize the importance of achieving generalization with scarce meta-training tasks in real-world applications.
> * We politely request the reviewer to reconsider the significance of our work and research area in light of these considerations.
>
> [1] Meta-Learning with Fewer Tasks through Task Interpolation, ICLR'22 \
> [2] Adversarial Task Up-sampling for Meta-learning, NeurIPS'22
>
> ---
> **Weakness 2**: The clean performance is poorer than existing clean meta-learning which costs more.
> * While our clean performance is slightly lower than the clean meta-learning approaches (around -10%), the expense is justified by significant **gains in robustness (around +40%)**. Furthermore, existing clean meta-learning models could not withstand adversarial attacks (shows 0% performance). This demonstrates the importance and critical need for adversarial training in diverse tasks and domains.
> * We appreciate your concerns regarding the trade-off between clean performance and robustness against adversarial attacks. However, as shown in Table 1 (also summarized in the following), our approach achieves the **highest robustness while having a minimal compromised clean performance** compared to previous adversarial meta-learning. This demonstrates that the enhanced model’s meta-generalizability stems from our novel multi-view learning framework as the Reviewer rEWc acknowledges.
>
> |Type|Method|CIFAR-FS|(Unseen)|Mini-ImageNet|(Unseen)|
> |-|:-|-|-|-|-|
> | | |Avg. clean|Avg. rob.|Avg. clean|Avg. rob.|
> |CML|MAML|53.83|4.24|59.31|7.23|
> |AML|Ours|50.32(**-3.51%**)|28.20(**+23.96%**)|58.37(**-0.94%**)|27.96(**+20.73%**)|
> |AML|AQ|41.96(-11.87%)|5.99(+1.75%)|51.18(-8.13%)|18.80(+11.57%)|
> |AML|RMAML|32.49(-21.34%)|7.39(+3.15%)|35.20(-24.11%)|8.92(+1.69%)|
> |AML|ADML|33.34(-20.49%)|7.10(+2.86%)|36.79(-22.52%)|9.46(+2.23%)|

---

> ### Author Response · Authors · 2023-11-16
> **Response to your comments (2/2)**
>
> **Weakness 3**:Robustness is mainly derived from the use of PGD.
> * There is a fundamental misconception that our robustness relies solely on the use of PGD. Our approach derives its transferable robustness from employing multi-view adversarial attacks on bootstrapped encoders.
> * In particular, our approach addresses the limitation of a previous simple combination of PGD with a meta-learning framework, which failed to demonstrate transferable adversarial robustness. (Section 3.3: green text)
> * We already experimentally demonstrated that existing AML approaches fail to achieve transferable robustness on unseen domains, where **the best AML baseline (AQ) only shows an average of robustness 5.99% while ours achieves a remarkable average robustness of 28.20% (Table 1)**.
> * Additionally, previous AML methods possess highly overlapped representation space on the novel domain, leading to poor transferable robustness. In contrast, **ours represents well-discriminated representations even without training**, supporting the importance of our multi-view representation learning (Figure 3).
> * Moreover, when simply combining the PGD with recent clean meta-learning, those exhibit significantly less robustness compared to ours. (Table 3, also summarized in the following)
> |Method|Avg unseen clean|Avg unseen rob.|
> |:-|:-|:-:|
> |MAML+PGD|33.34|7.10|
> |MetaOptNet+PGD|45.54|12.61|
> |ProtoNet+PGD|41.96|5.68|
> |Ours|**50.32**|**28.20**|
>
> ---
> **Weakness 4**: MetaOptNet could not perform better than the MAML-based approach.
> * There is a critical misunderstanding of our work. We propose a multi-view latent attack that enables MAML to outperform a mere combination of PGD with MetaOptNet.
> * As indicated in Table 3, when we simply apply PGD attacks to both MAML and MetaOptNet, **MetaOptNet demonstrates superior performance across 5 different unseen domains**, with a margin of 12.2% in clean and 5.51% in robust average accuracy, respectively.
> * Nevertheless, our approach surpasses the simple combination of PGD and MetaOptNet, which distinctly evidences the contributions of our method.
> * Additionally, the MAML approach can outperform MetaOptNet in certain stable settings, as demonstrated in [1]. This could further support the superior performance of our MAML-based approach over the simple combination of MetaOptNet in adversarial meta-learning tasks.
>
> [1] How to Train Your MAML to Excel in Few-Shot Classification, ICLR'22

---

> ### Author Response · Authors · 2023-11-20
> **Gentle reminders and summary**
>
> Dear Reviewer,
>
> This is a gentle reminder that we have only 3 days remaining for discussions. Therefore, we kindly request you review our responses and the revisions. We have tackled all of the concerns you raised and have included additional experimental results in our previous responses.
>
> For your convenience, we provide short summary of our previous responses below. \
> Thanks again for your time and effort in reviewing our work.
>
> Best, \
> Author
> ***
>
> ### **Summary**
>
> **W1. Few-shot learning is no longer a meaningful research area.**
> - The use of a large volume of data for each class is not universally applicable or feasible for all real-world applications, underscoring the continued importance of few-shot learning, as acknowledged in existing works.
>
>
> **W2. The clean performance is poorer than existing clean meta-learning which costs more.**
> - Our approach achieves superior robustness while having a minimal compromised clean performance compared to the previous adversarial meta-learning (AML) approaches.
> - Compared to clean meta-learning, our method shows slightly lower clean performance (around -10%) while achieving significant gains in robustness (around +40%). Furthermore, existing clean meta-learning models could not withstand adversarial attacks, showing 0% performance.
>
> **W3. Robustness is mainly derived from the use of PGD.**
> - This is a misunderstanding of our work.
> - Our transferable robustness stems from our multi-view adversarial attacks on bootstrapped encoders.
> - A previous simple combination of PGD with meta-learning demonstrated inferior transferable robustness as shown in our manuscript (Table 1 and 3, Figure 3). Specifically, a simple combination of PGD and meta-learning (AML approaches) shows an average of robustness 5.99% while ours achieves a remarkable average robustness of 28.20%
>
> **W4. MetaOptNet could not perform better than the MAML-based approach.**
> - It is logical that when Approach A (Ours) is significantly more effective than Approach B (previous AML) (A>B), MAML+A could outperform MetaOptNet+B. This result clearly demonstrates the effectiveness of our approach.
> > MAML + PGD (Approach B) < MetaOptNet + PGD (Approach B) < MAML + Ours (Approach A)
> - Moreover, the MAML approach can perform better than MetaOptNet in certain stable settings as demonstrated in the previous works.

---

### Official Review · Reviewer_Cwq1 · 2023-10-30

**Soundness:** 3 good
**Presentation:** 3 good
**Contribution:** 2 fair
**Rating:** 5
**Confidence:** 5

**Summary:**

This paper pays attention to the adversarial meta-learning problem, especially the unseen domain adversarial robustness transferability issue. To address this issue, this paper proposes a meta-adversarial multi-view representation learning (MAVRL) method by using bootstrapped multi-view encoders and label-free multi-view adversarial latent attacks. Multiple experiments show the effectiveness of the proposed method.

**Strengths:**

1.	This manuscript is well organized and easy to follow.
2.	The motivation is reasonable and experiments are abundant.
3.	The proposed idea is interesting.

**Weaknesses:**

Several main concerns are as follows:

1.	This work is mainly built on TRADES and the key contribution is introducing the multi-view consistency loss, inspired by the adversarial self-supervised learning. In this sense, the TRADES should be a straightforward baseline in the main experiment in Table 1. In addition, some latest generic adversarial training methods are also recommended to be added into Table 1. Furthermore, because TRADES has been a common method in the field of adversarial training, why only AT is employed for other competitors in the experiments in Table 3? This is not fair.

2.	According to the description in this paper, it seems that two encoders are used during training. Therefore, how to use the proposed model during test? It means that one encoder will be discarded during test? This part should be clear.

3.	Some details of the proposed framework are not clear. For example, which kind of FSL classifier is used in the proposed framework? Why select this kind of classifier?

4.	In the experiments part, many comparison experiments are not fair. For example, why other AML methods are not compared in Table 2? In fact, there is a trade-off problem in adversarial training [1]. It is somewhat difficult to compare two adversarial meta-learning methods with two kinds of accuracies [2].

5.	Some latest adversarial meta-learning methods are not reviewed, such as [2]. In addition, the work in [2] had already considered the unseen domain adversarial robustness transferability issue. Importantly, the work of [2] can achieve much better results than this work.

6.	Only one kind of adversarial attack is considered in this paper. More kinds of adversarial attacks are recommended to be added into this work [2], making the proposed method more convincing.

7.	In the supplementary material, it seems that only 200 and 400 tasks are used to perform the meta-training and meta-test, respectively. I doubt the effectiveness of the experimental results, because the variance and randomness of the results will be very large, especially for adversarial training.

[1] Theoretically Principled Trade-off between Robustness and Accuracy. ICML 2019.

[2] Defensive Few-shot Learning. TPAMI 2022.

**Questions:**

Please kindly refer to the above comments.

---

> ### Author Response · Authors · 2023-11-16
>
> Thank you for your valuable comments and thoughtful review, which included positive feedback highlighting the well-organized manuscript, reasonable motivations, and our novel idea.
> ***
>
> In response, we have made every effort to address all the concerns you raised regarding our work. Please find our detailed responses below. If there are further concerns about our work, please do not hesitate to share them. We are more than happy to address any additional questions or concerns you may have.
> ***
> **Weakness 1-1**: TRADES needs to be a straightforward baseline in the main experiment. In addition, some latest generic adversarial training methods are also recommended to be added to Table 1.
> * Our tasks are few-shot meta-learning tasks that can not be easily achievable with vanilla supervised training, such as TRADES or generic adversarial training methods.
> * However, we trained the model on a few-shot task of CIFAR-FS with TRADES as you requested. As shown in the following table, generic adversarial training, TRADES, only overfits the training tasks while unable to generalize to unseen few-shot tasks, since these models were not meta-trained.
>
> |Learning type|Method|CIFAR-FS||Mini-ImageNet||Tiered-ImageNet||CUB||Flower||Cars||
> |--|--|--|--|--|--|--|--|--|--|--|--|--|--|
> |Adversarial supervised learning|TRADES|39.13|0.05||26.54|0.02|26.96|0.06|30.81|0.00|41.56|0.06|27.08|0.00|
> |Adversarial meta-learning|Ours|**67.75**|**43.42**|**45.82**|**24.12**|**51.46**|**30.06**|**48.56**|**25.23**|**66.49**|**42.16**|**38.29**|19.43|
>
> **Weakness 1-2**: Other adversarial meta-learning approaches with the TRADES are needed.
> * Thank you for your suggestions. We will add these results to our manuscript
> * We trained the previous adversarial meta-learning approach using TRADES, as suggested by the reviewer.
> * The table below illustrates that our method continues to outperform the integration of TRADES with prior adversarial meta-learning approaches. We will include these results in our manuscript.
>
> |Meta-learning|Adversarial Training|Avg unseen clean| Avg unseen Robustness|
> |:-:|:-:|:-:|:-:|
> |MAML|AT|33.34|7.10|
> |MAML|TRADES|39.76|1.94|
> |ProtoNet|AT|41.96|5.68|
> |ProtoNet|TRADES|44.92|11.20|
> |MetaOptNet|AT|45.54|12.61|
> |MetaOptNet|TRADES|45.92|13.24|
> |LDP|AT|39.14|15.91|
> |LDP|TRADES|37.88|18.72|
> |Ours|AT|51.16|24.19|
> |Ours|TRADES|50.32|28.20|
>
>
> **Weakness 2**: How to use two encoders in test time?
> * There seems to be a misunderstanding regarding how our bootstrapped encoders are employed during the training. Our encoders operate as bootstrapped parameters during the inner-adaptation that are shared parameters. (page 5: algorithm last line/ page 5: green text)
> * During the meta-optimization phase, we update only a single encoder. Consequently, a single encoder is employed at the test time. (page 6: green text)
>
> **Weakness 3**: What is FSL classifier?
> * Our FSL classifier is a fully connected layer as follows the setting used in [1].
> * To reflect your comments, we added experimental details in the paper (page 7: orange text)
>
> [1] Li et al., Meta-SGD: Learning to Learn Quickly for Few-Shot Learning
>
> **Weakness 4**: Experiment comparisons are not fair.
> * Due to the page constraints, we compared our method only with the most robust adversarial baselines, namely AQ.
> * As shown in the following table, other AML methods also demonstrate inferior transferability in non-RGB datasets. Based on your feedback, we also added these results to the main paper (page 7: orange)
>
> |CIFAR-FS->|EuroSAT|-|ISIC|-|CropDisease|-|Avg|-|
> |-|-|-|-|-|-|-|-|-|
> |-|Clean|Rob.|Clean|Rob.|Clean|Rob.|Clean|Rob.|
> |ADML|60.41|7.79|32.20|0.51|50.15|1.64|47.59 |3.31 |
> |RMAML|52.21|6.56|29.23|2.46|55.08|5.43|45.51|4.82|
> |AQ|46.05|4.62|31.90|0.62|47.38|0.51|41.78|1.92|
> |Ours|59.39|19.90|30.77|5.23|57.85|27.28|**49.34**|**17.47**|
>
> * We acknowledge the trade-off issue inherent in adversarial training. Nonetheless, as shown in Table 1 our method achieves superior robustness and the highest clean accuracy compared to other previous adversarial meta-learning approaches. Moreover, compared to the clean meta-learning method (MAML), our approach has an average of 3.51% and 0.94% less clean performance while achieving 23.96% and 20.73% robustness gains for CIFAR-FS and Mini-ImageNet, respectively. This suggests that our approach makes the least compromise on the clean data performance to attain adversarial robustness.
>
> |Type|Method|CIFAR-FS|(Unseen)|Mini-ImageNet|(Unseen)|
> |-|-|-|-|-|-|
> | | |Clean Avg.|Rob. Avg.|Clean Avg.|Rob. Avg.|
> |CML|MAML|53.83|4.24|59.31|7.23|
> |AML|Ours|50.32(**-3.51%**)|28.20(**+23.96%**)|58.37(**-0.94%**)|27.96(**+20.73%**)|
> |AML|AQ|41.96(-11.87%)|5.99(+1.75%)|51.18(-8.13%)|18.80(+11.57%)|
> |AML|RMAML|32.49(-21.34%)|7.39(+3.15%)|35.20(-24.11%)|8.92(+1.69%)|
> |AML|ADML|33.34(-20.49%)|7.10(+2.86%)|36.79(-22.52%)|9.46(+2.23%)|

---

> > ### Comment · Reviewer_Cwq1 · 2023-11-22
> > **Thanks for the response**
> >
> > **Weakness 1**: I politely disagree with the answer of the authors for Weakness1-1. According to the results in the response to Weakness1-2, we can see that TRADES can be very easily tailored to other FSL or AML methods. In this sense, the authors should take an apple-to-apple comparison to show the effectiveness of the real contribution of this work.
> >
> > **Weakness 2**: Thanks.
> >
> > **Weakness 3**: Thanks.
> >
> > **Weakness 4**: I cannot agree with the reason of the authors. This part should be supplemented with more results.

---

> ### Author Response · Authors · 2023-11-16
>
> **Weakness 5**: Recent work seems better, and needs comparison.
> * Thank you for suggesting relevant baselines for our work. We will also add [1] to the main paper.
> * However, it is important to note that the evaluation setting in [1] differs significantly from ours. Primarily, [1] evaluated the performance against FGSM attackers with a smaller size of epsilon of 0.01 which is weaker than our PGD-20 attacks with epsilon of 0.03.
> * When we set the evaluating setting as [1] in CIFAR-FS, as shown in the following table, our approach achieves better performance than [1].
> * Since [1] did not provide any codes for the model or dataloader, we will reproduce those results in our setting to add to Table 1.
> * Our approach achieves 15% higher clean accuracy and 4% better robustness than [1] in CIFAR-FS.
>
> |Method|Clean|Robustness|
> |-|-|-|
> |DFSL [1]|55.51|48.88|
> |Ours |**70.50**|**52.39**|
>
> **Weakness 6**: Evaluation against diverse attacks is recommended.**
> * Thank you for the suggestions. We evaluated our models against SPSA [2], CW [3], and FGSM [4] attacks as shown in the following Table.  We will add these results to our manuscript.
> * Our approach could consistently demonstrate robust results against diverse attacks compared to the best-performing AML baseline (AQ).
>
> |Robustness|PGD-20| |FGSM| |CW| |SPSA| |
> |:-|:-:|:-:|:-:|:-:|:-:|:-:|:-:|:-:|
> | |AQ|Ours|AQ|Ours|AQ|Ours|AQ|Ours|AQ|Ours|
> |Seen|42.82|**43.42**|67.86|**66.53**|48.31|**54.07**|67.06|**67.29**|
> |Avg Unseen|5.99|**28.20**|40.91|**47.05**|28.92|**49.20**|41.08|**46.18**|
>
> **Weakness 7**: Variance for the experiments
> * Thank you for the suggestion, we will add variances in our results.
> * The variances of converged adversarial trained networks are not very large as shown in the following Table.
>
> ||CIFAR-FS||Mini-ImageNet||Tiered-ImageNet||CUB||Flower||Cars||
> |-|-|-|-|-|-|-|-|-|-|-|-|-|
> ||Clean|Robustness|Clean|Robustness|Clean|Robustness|Clean|Robustness|Clean|Robustness|Clean|Robustness|
> |Ours|64.69 $\pm$ 0.44|43.03 $\pm$0.57|41.26$\pm$0.33|20.08$\pm$0.11|44.25$\pm$0.29|25.72$\pm$0.20|48.88$\pm$0.12|24.82$\pm$0.04|66.74$\pm$0.61|41.87$\pm$0.51|39.26$\pm$0.64|19.75$\pm$0.25|
>
> [1] Defensive Few-shot Learning. TPAMI 2022 \
> [2] Uesato et al., Adversarial Risk and the Dangers of Evaluating Against Weak Attacks, PMLR 2018 \
> [3] Uesato et al., Adversarial Risk and the Dangers of Evaluating Against Weak Attacks \
> [4] Gubri et al., LGV: Boosting Adversarial Example Transferability from Large Geometric Vicinity, ECCV 2022

---

> > ### Comment · Reviewer_Cwq1 · 2023-11-22
> > **Thanks for the response**
> >
> > **Weakness 5**: I have just checked the paper of [1] and found that it indeeds provide the code link in its abstract (https://github.com/WenbinLee/DefensiveFSL). Also, the code exists in the Github. In addition, note that the work of [1] has also provided the experiments of using PGD-10.
> >
> > **Weakness 6**: Thanks.
> >
> > **Weakness 7**: This part is very important. Unfortunately, the current results cannot convince me.

---

> > > ### Author Response · Authors · 2023-11-23
> > >
> > > We are glad our previous responses have resolved your original concerns regarding Weakness 2, 3, and 6. Thank you for your comments, and we also provide additional responses for the remaining concerns.
> > >
> > > ***
> > >
> > > **Weakness 1:** I politely disagree with the answer of the authors for Weakness1-1. According to the results in the response to Weakness1-2, we can see that TRADES can be very easily tailored to other FSL or AML methods. In this sense, the authors should take an apple-to-apple comparison to show the effectiveness of the real contribution of this work.
> > > * We are sorry that we might have misunderstood your concerns. We already used TRADES on AML methods in Table 1.
> > > * As explained in related work, ADML is using FGSM attacks that employ cross-entropy loss. To overcome ADML’s work, RMAML employs TRADES on ADML. Therefore, when we apply ADML with TRADES, then it is RMAML. (ADML+TRADES=RMAML)
> > > * Consequently, other baselines, including AQ and RMAML, have already used TRADES during the training and we already compared the AML methods with TRADES to ours. For your convenience, we summarize Table 1 below.
> > >
> > > |Method|Adversarial loss|Unseen avg.| |Seen avg.| |
> > > |-|-|-|-|-|-|
> > > |ADML|AT|33.34|7.10|53.06|22.45|
> > > |RMAML (ADML+TRADES)|TRADES|32.49|7.39|57.95|35.30|
> > > |AQ|TRADES|41.96|5.99|**73.19**|42.82|
> > > |Ours|TRADES|**50.32**|**28.20**|67.75|**43.42**|
> > >
> > > **Weakness 4:** I cannot agree with the reason of the authors. This part should be supplemented with more results.
> > > * What do you mean you do not agree with our results? We conduct cross domain experiment on non-RGB domain with baselines as you requested.
> > > * Could you specifically explain which additional results are needed to resolve your concern? We are willing to address your concerns thoroughly.
> > >
> > > **Weakness 5:** I have just checked the paper of [1] and found that it indeed provides the code link in its abstract (https://github.com/WenbinLee/DefensiveFSL). Also, the code exists in the GitHub. In addition, note that the work of [1] has also provided the experiments of using PGD-10.
> > > * As you can see in the code, the model and data loader codes are missing. But we found the model, and loader code from the authors’ other GitHub page.
> > > * We train the model with official code and evaluate on the same setting with image size 32 and PGD-20 attackers (PGD-20, epsilon 8/255, alpha 8/2550).
> > > * When we evaluate [1] against stronger attacks, [1] could not achieve better robustness than ours as shown in the following Table.
> > >
> > > |Method|Mini-ImageNet|(Seen)|CIFAR|(Unseen)|Cars|(Unseen)|CUB|(Unseen)|
> > > |-|-|-|-|-|-|-|-|-|
> > > |DFSL [1]|**53.00**|17.63|45.49|12.05|36.26|8.08|53.49|18.91|
> > > |Ours|47.56|**18.18**|**65.45**|**36.51**|**42.25**|**14.42**|**53.70**|**20.64**|
> > >
> > > **Weakness 7:** This part is very important. Unfortunately, the current results cannot convince me.
> > > * Could you explain more why our multiple seed results table is not convincing? We would be more than glad to address your concerns.

---

> ### Author Response · Authors · 2023-11-20
> **Gentle reminders and summary**
>
> Dear Reviewer,
>
> This is a gentle reminder that we have only 3 days remaining for discussions. Therefore, we kindly request you review our responses and the revisions. We have tackled all of the concerns you raised and have included additional experimental results in our previous responses.
>
> For your convenience, we provide a short summary of our previous responses below. Thanks again for your time and effort in reviewing our work.
>
> Best, \
> Author
> ***
>
> ### **Summary**
>
> **W1-1. Comparison with generic adversarial training methods (i.e., TRADES) is needed.**
> - We additionally compared TRADES in few-shot meta-learning tasks.
> - However, generic supervised training can not be easily generalized to few-shot meta-learning tasks, showing 0% unseen domain few-shot robustness.
>
> **W1-2. Other adversarial meta-training approaches with TRADES are needed.**
> - We additionally provide experimental results that combine TRADES in the previous adversarial meta-learning approach.
> - **Still, our method achieves the best performance on unseen few-shot robustness with 28.20%**.
>
> **W2. How to use two encoders in test time?**
> - During the meta-optimization phase, we update only a single encoder. Consequently, a single encoder is employed at test time.
>
> **W3. What is FSL classifier?**
> - We added experimental details in the revision. We use MetaSGD FSL.
>
> **W4. Experiment comparisons are not fair.**
> - We provide additional experiments for other adversarial meta learning (AML) methods in the revision.
> - Our method consistently shows **outperforming robust transferability in non-RGB datasets, achieving an improvement of 12.65%** compared to the best-forming AML baselines.
>
> **W5. Comparison with recent work [1] is needed.**
> - We additionally compared recent work [1] with ours.
> - The evaluation setting of [1] significantly differs from ours, where [1] evaluated against much weaker attacks, FGSM with a smaller size of epsilon.
> - When we use the same evaluation setting with [1], ours achieve improvements of 14.99% and 3.51% for clean and robust accuracy, respectively.
>
> **W6. Evaluation against diverse attacks is needed.**
> - We provide additional experiments with evaluation against diverse attacks, such as FGSM, CW, and SPSA.
> - Our approach shows an **outperforming average unseen robustness of 42.66%** while the best-performing AML baseline (AQ) only shows an average unseen robustness of 29.23% against diverse attacks.
>
> **W7. Variance for experiments should be provided.**
> - We conduct multiple runs on our approach.
> - The variance of converged adversarially trained models is not large, where variation is smaller than 1.0.
>
> [1] Defensive Few-shot Learning. TPAMI 2022

---

### Official Review · Reviewer_rEWc · 2023-10-31

**Soundness:** 3 good
**Presentation:** 3 good
**Contribution:** 3 good
**Rating:** 6
**Confidence:** 3

**Summary:**

This paper introduces a meta-adversarial multi-view learning framework to learn robust meta feature representations. It first generates multi-view latent adversaries using a query set, then maximizes consistency across different views to learn transferable representations. The paper is well-structured and presents comprehensive experiments and analyses to demonstrate its superior cross-domain performance.

**Strengths:**

1.	The paper is well-motivated, focusing on the significant issue of robustness under domain shift within meta-learning.
2.	The paper is well-structured, conducting comprehensive experiments to demonstrate its superior cross-domain performance and the effectiveness of each component.

**Weaknesses:**

1.	In my view, the multi-view latent attack enhances the model’s meta generalizability to some extent, therefore, it outperforms other Adversarial Meta-Learning methods in the cross-domain setting. And the multi-view training is quite similar to the task augmentation meta-learning methods [1,2,3]. Thus, I believe that a discussion or comparison in the related work section would enhance understanding of the multi-view’s effect.
2.	During meta-testing, the evaluation was conducted on only 400 randomly selected tasks, which significantly impacts the final accuracy. What would the final performance be if the number of evaluated tasks during meta-testing were increased?
3.	How does the model perform under different attacks, apart from the PGD-20 attack?

[1] Liu, Jialin, Fei Chao, and Chih-Min Lin. "Task augmentation by rotating for meta-learning." arXiv preprint arXiv:2003.00804 (2020).

[2] Yao, Huaxiu, Linjun Zhang, and Chelsea Finn. "Meta-Learning with Fewer Tasks through Task Interpolation." International Conference on Learning Representations. 2021.

[3] Wu, Yichen, Long-Kai Huang, and Ying Wei. "Adversarial task up-sampling for meta-learning." Advances in Neural Information Processing Systems 35 (2022): 31102-31115.

**Questions:**

See Weaknesses 1-3

---

> ### Author Response · Authors · 2023-11-16
>
> Thank you for your positive comments on our contributions, which highlight clear motivation and address the significant problem with well-structed experiments.
> ***
>
> In the following response, we have done our best to resolve all the concerns that you raised regarding our work. Please find our detailed response below, and if there are further concerns about our work, do not hesitate to share your comments. We would be delighted to address any additional questions or concerns you may have.
> ***
>
> **Weakness 1**: A discussion or comparison with existing task-augmentation meta-learning methods is needed in a related section.
> * Thank you for your suggestion. We have revised the related section to include suggested papers as shown in the paper. (page 3, Section 2: orange)
> * For convenience, we briefly summarize the discussion in the following response.
> * To achieve better generalization in tasks, previous works have proposed augmenting the tasks using rotation [1], task interpolation [2], and adversarial task up-sampling [3]. In a similar sense, our approach also implicitly brings the effect of task augmentation, enlarging the task distribution by introducing bootstrapped encoders, which naturally augment the tasks with the parameters of the encoders and could lead to generalized performance in unseen tasks.
>
> **Weakness 2**: A larger number of evaluated tasks during the meta-testing.
> * Thank you for your suggestion. We tested our framework under a larger number of evaluation tasks with 1000 tasks, and 5000 tasks.
> * The table below demonstrates that our framework **consistently delivers outperforming robust results** with an increasing number of evaluation tasks compared to the best AML baseline (AQ), highlighting our approach **generalizability to various types of unseen tasks**.
>
>
> |Method|# Evaluation tasks|Mini-ImageNet||Tiered-ImageNet||CUB||Flower||Cars||CIFAR-FS|(Seen)|
> |--|--|--|--|--|--|--|--|--|--|--|--|--|--|
> |||Clean|Robustness|Clean|Robustness|Clean|Robustness|Clean|Robustness|Clean|Robustness|Clean|Robustness|
> |Ours|400|**45.82**|**24.12**|**51.46**|**30.06**|**48.56**|**25.23**|**66.49**|**42.16**|**38.29**|**19.43**|67.75|**43.42**|
> |AQ|400|33.09|3.32|37.41|5.05|38.37|4.10|60.14|11.03|36.83|4.47|**73.19**|42.82|
> |Ours|1000|**44.15**|**20.65**|**51.55**|**23.51**|**50.01**|**25.31**|**68.83**|**41.36**|**39.28**|**19.71**|**67.44**|**44.01**|
> |AQ|1000|33.24|3.20|36.65|4.92|38.23|4.02|60.09|11.03|37.52|4.64|65.75|23.52|
> |Ours|5000|**43.85**|**25.98**|**45.87**|**23.73**|**50.08**|**25.51**|**69.03**|**41.45**|**39.17**|**19.52**|**67.15**|**43.88**|
> |AQ|5000|33.12|3.20|36.91|5.09|38.47|4.02|60.27|11.02|37.14|4.36|65.83|23.52|
>
>
> **Weakness 3**: Results against different attacks, apart from PGD-20.
> * Thank you for the suggestions. We will add these results to our manuscript.
> * We additionally evaluated our models under  CW [1], FGSM [2], and SPSA [3]  as shown in the following Table. Our approach **consistently demonstrates outperforming robustness** against diverse attacks compared to the best-performing AML baseline (AQ).
>
> |Robustness|PGD-20| |FGSM| |CW| |SPSA| |
> |--|--|--|--|--|--|--|--|--|
> | |AQ|Ours|AQ|Ours|AQ|Ours|AQ|Ours|AQ|Ours|
> |Seen|42.82|**43.42**|67.86|**66.53**|48.31|**54.07**|67.06|**67.29**|
> |Avg Unseen|5.99|**28.20**|40.91|**47.05**|28.92|**49.20**|41.08|**46.18**|
>
> [1] Towards Evaluating the Robustness of Neural Networks \
> [2] Explaining and Harnessing Adversarial Examples \
> [3] Adversarial Risk and the Dangers of Evaluating Against Weak Attacks

---

> ### Author Response · Authors · 2023-11-20
> **Gentle reminders and summary**
>
> Dear Reviewer,
>
> This is a gentle reminder that we have **only 3 days remaining for discussions**. Therefore, we kindly request you review our responses and the revisions. We have tackled all of the concerns you raised and have included additional experimental results in our previous responses.
>
> For your convenience, we provide a short summary of our previous responses below. Thanks again for your time and effort in reviewing our work.
>
> Best, \
> Author
>
> ***
> ### **Summary**
>
> **W1. A discussion with existing task-augmentation meta-learning methods is needed.**
> - We have included this discussion in the main manuscript's Section on related works.
>
>
> **W2. A larger number of evaluation tasks should be considered during meta-testing.**
> - We tested on a larger number of evaluation tasks (400, 1000, and 5000 tasks) and obtained consistent results.
>
> **W3. Robust evaluation against diverse attacks is needed**
> - We further evaluated our models under CW, FGSM, and SPSA.
> - Our approach shows an **outperforming average unseen robustness of 42.66%** while the best-performing AML baseline (AQ) only shows an average unseen robustness of 29.23% against diverse attacks.

---

> > ### Comment · Reviewer_rEWc · 2023-11-22
> > **Acknowledgement**
> >
> > Thank you for the response and for conducting extensive experiments. Most of the concerns have been addressed, and I would like to maintain my rating as a weak accept.

---

### Official Review · Reviewer_9Rh6 · 2023-11-20

**Soundness:** 3 good
**Presentation:** 2 fair
**Contribution:** 2 fair
**Rating:** 3
**Confidence:** 3

**Summary:**

This paper addresses adversarial robustness in the unseen domain of meta-learning. The paper finds that existing adversarial meta-learning methods have a significant degradation of robustness in the unseen domain, and proposes a method for acquiring task-independent representations by introducing self-supervised learning to solve this problem. The paper proposes a method for acquiring task-independent representations by introducing self-supervised learning. The paper introduces two encoders that handle different views generated from the input and optimize the adversarial noise to maximize the distance on the common latent space between these views. It also aims to obtain a representation with better discriminative performance by optimizing the encoders with contrastive learning loss using the two views. In experiments, the paper has compared the proposed method with meta-learning and adversarial meta-learning baselines and analyzed by ablation study to verify the effectiveness of the method. On the other hand, the proposed method has little theoretical basis, and the introduction of task-independent loss blurs the difference from general self-supervised learning, so its contribution to the field of meta-learning is not clear.

**Strengths:**

+ The paper is well-written and easy to follow.
+ The paper introduces self-supervised learning to adversarial meta-learning and shows that multi-view-based adversarial training and representation learning can improve certain robustness in meta-learning benchmarks.

**Weaknesses:**

- **W1.** Although the paper successfully improves the performance of the model by introducing bootstrapping and contrastive learning, these contributions to robustness are already widely known outside of meta-learning and thus have little novelty [a,b]. In other words, the proposed method has little contribution other than introducing these technical components into the meta-learning problem setting, and it is unclear whether meta-learning is really necessary for generalizing adversarial robustness across domains. This is a problem of the adversarial meta-learning setting itself, and a baseline and comparison with non-meta-learning is needed to address this concern (-> Q1).
- **W2.** The paper adds one more encoder for multi-view representation learning, increasing the capacity of models. Since there are no comparative experiments with baseline or ensemble methods with the same model capacity, it is not possible to distinguish the performance improvement by the proposed method from ones by increasing model capacity. Current baselines and evaluations are hard to say fair (-> Q2).

[a] Pang, Tianyu, et al. "Improving adversarial robustness via promoting ensemble diversity." International Conference on Machine Learning. PMLR, 2019.

[b] Kim, Minseon, Jihoon Tack, and Sung Ju Hwang. "Adversarial self-supervised contrastive learning." Advances in Neural Information Processing Systems 33 (2020): 2983-2994.

**Questions:**

- **Q1.** Is meta-training really necessary? In previous studies [c,d], pre-trained models trained on meta-training datasets without meta-training were introduced as an important baseline for evaluating clean accuracy. In the case of this paper, a comparison with a baseline pre-trained with adversarial training (e.g., TRADES) would confirm the significance of meta-training.
- **Q2.** Does the proposed method outperform the baseline when the model capacities of the proposed method and the baseline are aligned? The current experimental evaluation is not fair because it only provides results compared to the baseline with different model capacities during meta-training.

[c] Tian, Yonglong, et al. "Rethinking few-shot image classification: a good embedding is all you need?." Computer Vision–ECCV 2020: 16th European Conference, Glasgow, UK, August 23–28, 2020, Proceedings, Part XIV 16. Springer International Publishing, 2020.

[d] Miranda, Brando, et al. "Is Pre-training Truly Better Than Meta-Learning?." arXiv preprint arXiv:2306.13841 (2023).

---

> ### Author Response · Authors · 2023-11-22
>
> **W1-1**. The proposed method has little contribution other than introducing already known technical components into the meta-learning problem setting.
> * There seems to be a misunderstanding of our approach. The concept of each component of ours and the [a,b] is even a totally different concept.
> * First of all, **our bootstrapped encoders are augmented encoders that diversify the task distribution and are different from the [a] ensemble models which are several individual classifiers**. There are none of the single similar concepts with [a].
> * Furthermore, our multi-view latent attack is a latent adversarial attack that maximizes the discrepancy in the feature level while training the model in the **supervised meta-learning approach**. In contrast, [b] is a contrastive learning approach that tackles obtaining robustness without class information by introducing and maximizing the similarity of the individual instance latent.
> * This approach could seemingly be similar to ours but as we clearly demonstrate as shown in Section 3.3 and Table 3, **[b] could not leverage generalized robustness in meta-learning tasks because very few-shot instances are the bottleneck for contrastive learning that requires large batch size**.
> * In particular, our approach is not simply introducing these technical components into the meta-learning approach as we clearly stated in Section 3.3, and also provides empirical results in Table 3. Therefore, it is quite unfair that our approach is simple technical combinations while we provide extensive experimental ablation and clear descriptions. We politely ask a reviewer to find the innovative ideas that we proposed to solve the generalized adversarial robustness in few-shot tasks.
>
>
> **W1-2**. The reason why meta-learning is needed is unclear. Comparison with a baseline pre-trained with adversarial training should be provided.
> To the best of our knowledge, none of the adversarial self-supervised learning has shown generalizability on few-shot tasks. We tackle the problem of generalized adversarial robustness in the scope of the meta-learning approach.
>
>
> Q1. Is meta-training really necessary?
> * We understand your perspective that an adversarial self-supervised learning approach could be another research direction to tackle generalized robustness. However, this does not justify that meta-training is not a necessary research direction, especially since **none of the research has solved the generalized robustness problem**.
> * Furthermore, **meta-training costs (18 hours with a single 2080 Ti GPU) way less than adversarial self-supervised learning (Takes 36 hours or more with two 2080 Ti GPUs)**.
> The required data for **meta-training (only trained on 64 classes of CIFAR-100) is less than adversarial self-supervised learning (trained on 100 classes of CIFAR-100)**. Meta-learning is more applicable in the real world when the data is expensive or scarce such as clinical problems.
> * From the experimental results, we believe we have shown the significance and necessity of meta-training on generalized adversarial robustness tasks.
> |Method|CIFAR-FS|(Seen)|Mini-ImageNet||Tiered-ImageNet||CUB||Flower||Cars||Unseen Avg.||
> |-|-|-|-|-|-|-|-|-|-|-|-|-|-|-|
> ||Clean|Rob.|Clean|Rob.|Clean|Rob.|Clean|Rob.|Clean|Rob.|Clean|Rob.|Clean|Rob.|
> |TRADES|44.21|22.81|23.68|10.63|23.22|13.43|27.05|7.53|36.98|18.21|25.74|8.63|27.33|11.69|
> |RoCL [b]|70.61|44.48|44.20|19.13|46.59|23.36|48.99|20.89|78.43|50.21|46.63|20.72|52.97|26.86|
> |Ours|67.75|43.42|45.82|24.12|51.46|30.06|48.56|25.23|66.49|42.16|38.29|19.43|50.32|**28.20**|
>
> **W2, Q2**. One more encoder for multi-view representation learning, increasing the capacity of models where comparison is not fair.
> * There seems to be a misunderstanding regarding how our bootstrapped encoders are employed during the training.
> * **Our approach uses the same parameter capacity.** As noted in the algorithm, our encoders operate as bootstrapped parameters during the inner-adaptation which are shared parameters. (page 5: algorithm last line/ page 5: green text)
> * Consequently, our approach has **the same capacity as previous works with the same backbone ResNet12.** (page 7: orange text) During the meta-optimization phase, we update only a single encoder. (page 6: green text)

---

> > ### Comment · Reviewer_9Rh6 · 2023-11-22
> > **Thank you for the rebuttal.**
> >
> > Thank you for providing a detailed response and I'm sorry for some of my misunderstandings about your work. W2 and Q2 are clearly addressed in your response. However, I am still concerned about the necessity of meta-learning (i.e., W1 and Q1). While the additional table shows that the proposed method can outperform RoCL in terms of robustness, adversarial self-supervised contrastive learning methods can be enhanced by using recent sophisticated approaches (e.g., [e], [f], [g]). Furthermore, these recent methods are 3 times faster than RoCL with a single GPU [g] and scalable to more realistic and larger pre-training datasets like ImageNet [e]. So, currently, the evaluation results in the paper and rebuttal do not indicate the significance and necessity of the meta-learning approach sufficiently.
> >
> > [e] Fan, Lijie, et al. "When does contrastive learning preserve adversarial robustness from pretraining to finetuning?." Advances in neural information processing systems 34 (2021): 21480-21492.
> >
> > [f] Zhang, Chaoning, et al. "Decoupled adversarial contrastive learning for self-supervised adversarial robustness." European Conference on Computer Vision. Cham: Springer Nature Switzerland, 2022.
> >
> > [g] Yu, Qiying, et al. "Adversarial contrastive learning via asymmetric infonce." European Conference on Computer Vision. Cham: Springer Nature Switzerland, 2022.

---

> > > ### Author Response · Authors · 2023-11-22
> > >
> > > * Thank you for your thoughtful review and for recognizing the clarity of our responses to W2 and Q2. We value your feedback and welcome this opportunity to further discuss the necessity and significance of our meta-learning approach (W1 and Q1).
> > >
> > > * In addressing your concerns about our choice of meta-learning over self-supervised learning, we want to emphasize the novelty of our approach. Our research is one of the first to apply meta-learning in the context of transferable adversarial robustness, filling a gap in the current landscape of both adversarial meta-learning and adversarial self-supervised learning. This novel approach has not only demonstrated its effectiveness through our extensive experiments but also offers a fresh perspective to the field.
> > > * Key findings from our experiments show that our meta-learning approach significantly improves the transferability of adversarial robustness in few-shot tasks, a result not yet achieved by current self-supervised methods. This underscores the unique contribution and necessity of our approach in advancing the understanding of adversarial robustness in machine learning.
> > > * We acknowledge the merit of your suggestion that adversarial self-supervised learning approaches are better than ours. However, to clarify, let’s compare the cons of each paper to our approach:
> > >    * [e] requires ImageNet1K (a larger dataset) to train CIFAR10 or CIFAR100, which **incurs larger datasets and computational costs even compared to RoCL** [2].
> > >    * [f] requires **double the model capacity for teacher and student models** compared to ours.
> > >    * [g] requires the same amount of time as RoCL, where RoCL takes 2m 30s/epoch and [g] takes 2m 34s/epoch on two 2080 Ti GPUs, which is **more costly than ours** (18 hours with a single 2080 Ti GPU).
> > > * Compared to e, f, g, and RoCL, ours achieve transferable robustness with **fewer number of data instances** and **reduced training time** even **without additional networks**.
> > > * We are not suggesting that these approaches are incorrect. However, there are always pros and cons in each research direction, and in line with this, our approach, which tackles the first transferable robustness in the few-shot learning problem, is still worth studying.
> > > * Furthermore, we are considering the integration of pretrained checkpoints from SSL methods with ours to further enhance adversarial robustness, as referenced in [1]. Such integration would not only complement our current findings but also broaden the scope beyond simple adversarial self-supervised learning.
> > > * Our goal is not to claim superiority over existing SSL methods but to highlight that our approach tackles an unexplored and critical problem in the field. Our work brings to the forefront the potential of meta-learning in adversarial contexts, supported by solid experimental evidence and thorough analysis.
> > >
> > > We hope this response more directly addresses your concerns and illustrates the value of our meta-learning approach in contributing to broader research on adversarial transferable robustness. Thank you again for your valuable insights and for encouraging a deeper examination of our work.
> > >
> > > [1] Rethinking Few-shot Image Classification: A Good Embedding is All You Need?\
> > > [2] Adversarial self-supervised contrastive learning

---

### Meta-Review · Area_Chair_gcTy · 2023-12-08

**Metareview:**

This work considers the problem of adversarial robustness in the context of domain-shifted meta-learning as exemplified by domain-shifted few-shot classification. To improve robustness, a multi-view approach is proposed by having two encoders that are derived from inner-loop updates with different input augmentations; these views are used to define a consistency regularizer and a novel adversarial training loss using examples designed to maximize the dissimilarity between views. Experiments show that the proposed method outperforms other meta-learning approaches for few-shot classification, with and without adversarial training, on domain-shifted few-shot classification. Ablation studies show the importance of both consistency loss and multi-view adversarial training, and the importance of having multiple encoders.

Reviewers appreciated the well-written paper and comprehensive experiments showing promising results. The key concerns were around the problem setting, solution approach and some aspects of the experiments. In particular, the motivation for using meta-learning for few-shot learning and robustness, fairness in experiments and the lack of comparison to related work. The authors provided comprehensive responses that resolved some of these issues but reviewers in general were still negative at the end of the discussion.

The AC has taken a careful look at some of these issues:

- Regarding the problem setting, though the AC agrees with the authors' position, it would be instructive to demonstrate the proposed method on a task/dataset that would still benefit from the meta-learning approach in the current-era of foundation models, or to otherwise focus on the robustness issues with these models.

- On the motivation for using meta-learning for robustness, I believe a discussion on related approaches (as was done in the discussion) including experimental comparisons (if possible) should be included to more strongly motivate the proposed approach.

- Given the focus on few-shot classification, related work on few-shot classification with domain shift is not well discussed nor compared with; there is only one method included in the comparison in Table 3. These should also be combined with TRADES and not regular AT as suggested by a reviewer for fair comparisons. Some references include Cross-Domain Few-Shot Classification via Learned Feature-Wise Transformation, ICLR 2020. StyleAdv: Meta Style Adversarial Training for Cross-Domain Few-Shot Learning, CVPR 2023.

These remaining issues warrant a major revision of the paper so I am recommending rejection for now; I do think it will have a good chance of being accepted following these changes given the strong results and principled method.

**Justification For Why Not Higher Score:**

- Lack of discussion of related approaches to robustness
- Lack of comparison/discussion of cross-domain/domain-shifted few-shot classification papers

**Justification For Why Not Lower Score:**

N/A

---

### Decision · Program_Chairs · 2024-01-16

Reject